# Temporal and Spatial Patterns and a Space–Time Cluster Analysis of Foot-and-Mouth Disease Outbreaks in Ethiopia from 2010 to 2019

**DOI:** 10.3390/v14071558

**Published:** 2022-07-16

**Authors:** Fanos Tadesse Woldemariyam, Samson Leta, Zerihun Assefa, Etsegent Tekeba, Dereje Shegu Gebrewold, Jan Paeshuyse

**Affiliations:** 1Laboratory of Host Pathogen Interaction, Department of Biosystems, Division of Animal and Human Health Engineering, KU Leuven, 3001 Leuven, Belgium; samson.leta@aau.edu.et; 2Department of Biomedical Sciences, College of Veterinary Medicine and Agriculture, Addis Ababa University, Bishoftu P.O. Box 34, Ethiopia; zerihun.assefa@aau.edu.et; 3Epidemiology Directorate, Ministry of Agriculture, Livestock and Fisheries, Addis Ababa P.O. Box 62347, Ethiopia; lilytekeba@gmail.com; 4National Animal Health Diagnostic and Investigation Center, Sebata P.O. Box 04, Ethiopia; derejeshegu@yahoo.com

**Keywords:** cluster, Ethiopia, FMD, FMDV, outbreak, spatiotemporal

## Abstract

Foot-and-mouth disease (FMD) is an endemic disease in Ethiopia, although space–time cluster and monthly variation studies have never been assessed at national level. The current study aimed to identify the spatial and temporal distribution of FMD outbreaks in Ethiopia from national outbreak reports over a period of ten years from 1 January 2010 to 31 December 2019. To this end, a total of 376,762 cases and 1302 outbreaks from 704 districts were obtained from the Minister of Agriculture for analyses. In general, the dry periods, i.e., October to March, of the year were recorded as the peak outbreak periods, with the highest prevalence in March 2012. The monthly average and the outbreak trends over ten years show a decrease of outbreaks from 2010 to 2019. Decomposing the FMD outbreak data time series showed that once an outbreak erupted, it continued for up to five years. Only 12% of the reported outbreaks were assigned to a specific serotype. Within these outbreaks, the serotypes O, A, SAT-2, and SAT-1 were identified in decreasing order of prevalence, respectively. When a window of 50% for the maximum temporal/space cluster size was set, a total of seven FMD clusters were identified in space and time. The primary cluster with a radius of 380.95 km was identified in the southern part of Ethiopia, with a likelihood ratio of 7.67 (observed/expected cases). The third cluster, with a radius of 144.14 km, was identified in the northeastern part of the country, and had a likelihood ratio of 5.66. Clusters 1 and 3 occurred from January 2017 to December 2019. The second cluster that occurred had a radius of 294.82 km, a likelihood ratio of 6.20, and was located in the central and western parts of Ethiopia. The sixth cluster, with a radius of 36.04 km and a likelihood ratio of 20.60, was set in southern Tigray, bordering Afar. Clusters 2 and 6 occurred in the same period, from January 2014 to December 2016. The fourth cluster in northern Tigray had a calculated radius of 95.50 km and a likelihood ratio of 1.17. The seventh cluster occurred in the north-central Amhara region, with a radius of 97 km and a likelihood ratio of 1.16. Clusters 4 and 7 occurred between January 2010 and December 2013. The spatiotemporal and cluster analysis of the FMD outbreaks identified in the context of the current study are crucial in implementing control, prevention, and a prophylactic vaccination schedule. This study pointed out October to March as well as the main time of the year during which FMD outbreaks occur. The area that extends from the south to north, following the central highlands, is the main FMD outbreak area in Ethiopia.

## 1. Introduction

Foot-and-mouth disease (FMD) is a trans-boundary disease endemic to Sub-Saharan Africa, South Asia, the Middle East, and other parts of the world. FMD is the most contagious disease affecting cloven-hoofed domestic and wild animals [1]. Clinically, it is characterized by pyrexia, lameness, loss of appetite, the drooling of saliva, and vesicular lesions of the tongue, feet, and teats [2].

The causative agent of FMD is a small-sized virus with a single-strand positive sense RNA genome of approximately 8.5 kbp. This virus is grouped within the family of *Picornaviridae*, genus *Aphthovirus*, and has seven immunogenically distinct serotypes (O, A, C, SAT-1, SAT-2, SAT-3, and Asia1) [3,4]. Over 65 topotypes (geographically distinct variants) have been described so far [3,5,6,7]. Cross protection between serotypes has not been reported. As a result, infections or vaccination do not necessarily induce protective immunity to reinfection with other serotypes [3,8].

The worldwide prevalence of FMD ranges from disease-free to endemic areas. Most economically developed countries have already eradicated the disease. In Sub-Saharan Africa, Southern and Eastern Asia, and some South American countries, the disease is endemic [8].

Globally, there are seven groups of FMDV serotypes, called pools, that circulate in certain areas. Pools 1 and 2 (Southeast and Southern Asia) and pool 3 (Euro-Asia (including the Middle East) are the first three pools located in Asia and Euro-Asia. Three circulating FMDV serotypes (O, A, and Asia 1) make up these three pools. Pool 4 (the eastern part of Africa) consists of serotypes O, A, SAT-1, SAT-2, and SAT-3. Pool 5 (the western part of Africa) contains serotypes O, A, SAT-1, and SAT-2. Pool 6 (southern Africa) comprises only the SAT serotypes. In pool 7, in South America, only serotypes A and O circulate [9].

In Ethiopia, the disease has been endemic since it was first reported in 1957, and it affects different species of animals with different levels of prevalence [10]. In cattle, FMD seroprevalence ranges from 1.4 to 53.6% at the animal level and has up to 61% seroprevalence at the herd level [11,12,13,14,15]. In domestic small ruminants, the seroprevalence is 4 to 11%, and in ungulate wildlife, it is 30% [4]. There are four serotypes known to circulate in Ethiopia. Of these, serotype O is the dominant serotype, followed by serotypes A, SAT-2, and SAT-1 [16]. Serotype C has not caused any outbreaks in Ethiopia since 2004 and 2008 [17,18,19]. Ethiopia is in its preparatory phase to implement FMD control. This initiative was raised to avoid trade restrictions for live animals and their products on the international market. A country-wide distribution of the disease and multi-serotype occurrence of the diseases complicates any attempt to control FMD, especially in a resource-scarce country like Ethiopia [16]. The national control plan (stage zero/one transition), as part of the progressive control pathway (PCP) designed and recommended by the OIE/FAO/EUFMD, is one of the pathways to control FMD in Ethiopia. This type of control approach requires spatiotemporal knowledge about the distribution of the disease with respect to monthly, yearly, and geographical cluster analyses of disease outbreaks. To the best of our knowledge, there has only been one study conducted to determine the spatiotemporal distribution of FMDV in the Amhara region [20]; however, no study has assessed the monthly variations, space–time clusters, and geographical analysis of FMD covering the whole of Ethiopia. Therefore, this study aimed to identify the spatial and temporal distribution of FMD using outbreak data reported in Ethiopia over a 10 year period, from January 2010 to December 2019.

## 2. Materials and Methods

### 2.1. Study Location and Description of Data

Ethiopia is located in the Horn of Africa and covers an area of 1,126,829 km^2^ (https://www.nationsonline.org/oneworld/ethiopia.htm, accessed on 10 August 2021), with an estimated human population of 116,527,002 in 2021 (https://www.worldometers.info/world-population/ethiopia-population/, accessed on 15 August 2021).

The agroecology of Ethiopia varies from lowlands to highlands. The country ranges from 4535 m above sea level (mount Ras Dashen) to −130 m below sea level (Dallol). In Ethiopia, the livestock subsector contributes 19% of the national GDP, 45% of the agricultural GDP, and one fifth of the country’s export earnings [21]. In the lowlands, extensive livestock grazing systems are practiced by the pastoralists. In the highlands, besides crop-livestock production, semi-intensive and intensive livestock production systems with small to medium scale herds are practiced [22]. The pastoral extensive system and small-scale, semi-intensive livestock production system have a low level of awareness for biosecurity, lack of access to information on livestock production, management, and veterinary services. The national data on FMD outbreaks from 2010 to 2019 was obtained from the Ministry of Agriculture (MOA), Livestock Department, and Epidemiology directorate. The data includes the number of cases, animal species (cattle and goats/sheep), estimated outbreak dates, and locations (on a district level, 704 out of 779 districts). The outbreak data were compiled from the monthly reports obtained from the district veterinary clinics. Reported cases were clinically diagnosed by the district’s animal health professionals, and laboratory confirmation was done for very few cases at the National Animal Health Diagnostics and Investigation Center (NAHDIC). NAHDIC has assigned the confirmed outbreak data to a specific serotype using an FMD antigen enzyme linked immunosorbent assay (ELISA).

Topotypes of serotypes were obtained from the publicly available database Pirbright (https://www.wrlfmd.org/east-africa/ethiopia#panel-5482, accessed on 25 September 2021). These topotypes have been determined using representative outbreak samples sent from the NAHDIC to Pirbright (a world reference laboratory for FMD). An association of the outbreak report, serotypes, and topotypes, with the epidemics cycle of the disease, was performed.

### 2.2. Data Analysis

The Ethiopian administrative structure has frequently been subject to modification. As of January 2020, there are nine administrative regions or states composed of 80 zones (an administrative hierarchy between region and districts) and two administrative cities. These zones include about 779 districts and over 30,000 peasant associations. Each district is composed of peasant associations that are an aggregation of villages, being a group of three to five villages, although this structure is sometimes unclear. According to the Federal Democratic Republic of Ethiopia, the 10 administrative regions were based on ethnic settlements and two city administrations as multiethnic settlements [23]. The estimated livestock population of each of the ten regions is depicted in Table 1. The outbreak data on FMD were collected at the district level and reported to the Ministry of Agriculture. Here, yearly and monthly cumulative cases of FMD were calculated for the study period.

Outbreak information and livestock data were recorded in Microsoft Excel (version 14.0.4734.1000) and analyzed using STATA version 14 [24]. The average outbreaks per district, per month were plotted by dividing the monthly outbreak sum during that specific month to a specific district. The count of an outbreak for ten years was also plotted separately (Figure 1). The temporal and special distribution of outbreaks over a 10 year period is visualized in Figure 2.

To reduce random variation and to ease the detection of underlying trends, a monthly moving average was used [25]. The 12 month rolling average was calculated by summing the values for a 12 month time window and dividing this sum by 12. This was calculated for each consecutive month, shifting the 12 month window by one month. The seasonality in the number of FMD outbreaks in Ethiopia was also calculated using the ratio of trend method of seasonal indices calculation, where the increase or decrease of outbreaks for a particular month was obtained. FMD outbreaks were decomposed in a time series using the R software package TTR (Technical Trading Rules) [26]. This was used to estimate the trends component, seasonality component, and random component (irregularity) using a temporal additive model, assuming our data magnitude of seasonal fluctuation and variation around the trend cycle does not vary with the level of the time series (10 years). The formula used to calculate the trend cycle component, seasonal component, and random component was Y = S + T + R, where y represents the observed actual outbreak, S is seasonal effect, T is the trend cycle and, R is the random effect (https://otexts.com/fpp2/components.html, accessed on 15 August 2021).

The trend cycle was used to see the trend component, assuming there is a regular, periodic fluctuation in disease occurrence associated with periodic changes in the size of susceptible host populations and/or efficient host contact that is prone to recurrent epidemics or epidemic pulsation. The seasonal trend as a cyclical component was also evaluated when the disease fluctuation was associated with a particular season. The randomness pattern of monthly occurrence of FMD outbreak was verified by a randomness test [27]. The trend and random datasets were also evaluated by removing the seasonal influence by de-seasonalizing the trend [28]. In this analysis, the outcome variable was FMD outbreak, whereas year/month of outbreak was a predictor variable. The visualizations of outbreak locations and spatiotemporal clusters were generated on a map using QGIS [29] version number 3.4.4. 

SaTScan^TM^ (version 9.5 free available https://www.satscan.org/, accessed on 15 August 2021) was used to analyze the space–time clusters of outbreak data [30,31]. In this particular study, cases were first recorded at the district level and edited at the national level, though originally, the sources of the cases were at the peasant association level. The cases recorded at this level were at a specific location in the district. A space–time permutation model was used to assess the space–time cluster model because our data lacked the entirety of the population at risk in a given district [32]. The analysis on SaTScan^TM^ was performed by uploading the case and location data (centroid) as an input file, selecting the space–time, then space–time permutation model, and finally, the high-rate cluster to run the analysis. As a window of the scan statistic, an interval (in time), a circle, or an ellipse (in space), as a testimony of time and space, was used). Additionally, space–time clustering was also possible with circular geographical bases and heights that corresponded to time. The observed/expected cases were calculated in the circular window, which is movable across each centroid of districts, assuming they are randomly distributed in space. The clusters were identified by dividing the observed cases in the district by the population at risk as expected cases, with the assumption of no clustering of the null hypothesis. The spatial and temporal analysis was performed using a 50% window size. The optimal temporal aggregate period was set as three years; longer or shorter durations do not give a meaningful outcome [32]. A Monte Carlo simulation with 999 replications of the dataset under the null hypothesis was used to calculate the maximum likelihood ratio function. A *p*-value of less than 0.05 was considered statistically significant and used to reject the null hypothesis that the stated cases are randomly distributed in space.

## 3. Results

### 3.1. Case Distribution of FMD Infections by Year

A total of 376,762 cases (a specific animal with a suspected clinical sign of FMD) were reported over the time period between 2010 and 2019 from 704 districts of Ethiopia. From these cases, 81.5% (307,219) occurred in cattle and 18.5% (69,543) in sheep. In terms of year, 2011 had the highest number of FMDV cases reported and 2018 had the lowest (Table 2).

### 3.2. Temporal Distributions of FMD Outbreak by Year

In the last ten years, a total of 1302 outbreaks (two or more linked cases of the same illness or the situations where the observed number of cases exceeded the expected number or where a single case of disease was caused by a significant pathogen in a specific period) were reported from 704 out of the 779 districts. The highest number of outbreaks was reported in 2012 (*n* = 357), followed by 2017 (*n* = 216), whereas the lowest number of outbreaks was observed in 2018 (*n* = 27). The cumulative outbreak distribution by month was also seen to vary from month to month over this ten year window (Figure 1). March was ranked first, with 231 outbreaks reported, followed by January, November, October, February, and April, all of which had more than 100 outbreaks. May, June, July, August, and September were the months with the lowest numbers of outbreak reports (<80). In this study, the outbreak period ranges from October to April, with an average recorded number of 100 outbreaks per month over the time period studied (Figure 2).

As is depicted in Figure 1, the distribution of FMD outbreak occurrence was summarized in terms of months (overall monthly average) and frequency of occurrence in a specific month/district of a specific year. The outbreak in a month of a specific year/district varies from 0 to 96, as indicated. Similarly, the overall monthly average calculated for a specific month shows a variation from 0 to 5.8. The linear regression analysis trend (in this case, the outbreak number in a specific month is considered as a response variable, whereas the year is an explanatory variable) of an outbreak over ten years shows that there was a decrease from year to year (Figure 1).

Outbreaks were also summarized with respect to their reporting district in a specific year. As depicted in Figure 2A, 357 districts in 2012, 152 districts in 2017, and 147 districts in 2019 reported an outbreak of FMD. On the other hand, only 24 districts reported an outbreak in 2018. In addition, 2012, 2017, and 2010 had cumulative outbreak reports of 357,152, and 128 outbreaks for the ten years. The sum of the outbreak distribution and the trend were plotted for 12 months and for ten years. The two month moving average trend shows that there was an increase in trend from January to April, and then a decline until September, followed by another rise in October (Figure 2B). In the case of a two year moving-average trend, it showed an increment in 2011 and a decline in 2013, and it was raised in 2015, with another decline in 2018 (Figure 2C). The two month moving-average trend analysis, in this case, fits with the seasonal indices component indicated in Figure 3 and Figure 4. Similarly, the two year moving-average trend also fits with the trend component analysis of Figure 4.

The seasonality in the number of FMD outbreaks in Ethiopia was evident. Accordingly, plotting the monthly indices (the measure of how a particular season through a cycle compares with the average season of that cycle) showed that the dry period (October to March) was found to have a higher frequency of outbreak as compared to other seasons of the year. The largest seasonal indices reported were for the month of March (8.75) and the lowest was for May (−4.63), indicating the peak (March) and the lowest (May) outbreak numbers (Figure 3).

By decomposing the FMD outbreak time series, the trend cycle, seasonal (as a special cyclical trend), and random components were estimated. Here the trend cycle and seasonal trend were used, considering that the regular and periodic fluctuation of the epidemics of FMD incidence is related to a particular season. The seasonality (dry period of the year) in the number of outbreaks was apparent (upward wave), as observed in Figure 3 and Figure 4. The cyclical pattern, as shown in the trend cycle component in Figure 4, shows two cycles from 2010 to 2019. The first cycle is from 2010 to 2014, whereas the second cycle is from 2014 to 2018, each constituting five years. The two FMD outbreak cycles peak in 2012 and 2017, respectively, as indicated by the blue arrow in Figure 4.

### 3.3. Outbreak and Serotype Distribution

The country-wide distribution of outbreaks was evaluated, and it was found that the majority of the districts had at least one outbreak and very few of them had greater than five outbreak reports. The remaining districts had two to five outbreak reports. Approximately 90% (704 of 779 districts) of the country had experienced at least one outbreak (Figure 5). A total of 1302 outbreaks were reported in ten years’ time, and 84% of them were single outbreaks, while two, three, four, or five outbreak reports accounted for 12% of the report. Outbreaks with more than five outbreaks comprised only 4% of all the outbreaks reported.

From all the outbreak samples, so far, only 12% of them were assigned to specific serotypes by FMD antigen ELISA. The remaining outbreaks might not have been sampled, or they were sampled but not analyzed. Serotype O is widely distributed in the country, whereas serotypes A, SAT-2, and SAT-1 are located in the central part of the country. This indicates that the control strategy should be designed to deal with the regional serotype prevalence of FMD in Ethiopia (Figure 6). The viral protein 1 sequence-based topotypes (viruses adapted to one geographical area) reported in this period for serotype O were EA-3 (2010–2015, 2017–2019) and EA-4 (2013, 2016, and 2019); for serotype A, they were A/Africa G-IV (2015, 2017–2019) and A/Africa G-I (2018); and for serotype SAT-2, they were SAT-2/XIII (2010), SAT-2/VII-Alx-12 (2014–2015), and SAT-2/VII-Lib-12 (2018). For SAT-1, no sequence-based topotypes were reported other than those that were determined using the FMD Antigen ELISA for serotyping.

### 3.4. Space–Time Cluster Analysis

Using the space–time permutation scan statistic test, at a window set of 50%, a total of seven FMD outbreak clusters were identified in Ethiopia (Figure 7). The cluster numbering was based on the number of cases in that particular area as compared to other areas, considering that the spatial and temporal locations are independent. Accordingly, the result showed that the most likely primary cluster was the largest one based on its likelihood ratio test statistics. This cluster was primarily seen in the southern part of Ethiopia, crossing into Kenya and Somalia, with a radius of 380.95 km. The time frame included in this cluster was 1 January 2017 to 31 December 2019. The observed (2268.8) and expected (17,400) likelihood ratio of this cluster was 7.7, with a *p*-value of <0.0001.

SaTScan^TM^ also identified the second most likely cluster based on the likelihood ratio of test statistics and other clusters in different parts of the country. The time frame for the secondary cluster was 1 January 2014 to 31 December 2016, with observed and expected cases of 14,059 and 2266.83, respectively. The likelihood ratio (observed/expected) of this cluster was 6.2, with a *p*-value of <0.0001. This cluster was the second largest, with a radius of 294.8 km, and it encompassed the central-west and western parts of the country and jumped to south Sudan.

The third cluster was also identified in the northeastern part of the country. The time frame for the third cluster was 1 January 2017 to 31 December 2019, with observed and expected cases of 5583 and 986.8, respectively. The likelihood ratio (observed/expected) of this cluster was 5.7, with a *p*-value of <0.0001. The radius of this cluster was 144.1 km, encompassing the southern Amhara and northern Afar regions.

The fourth cluster was also identified in the northern part of the country, crossing to Eritrea. The time frame for the fourth cluster was 1 January 2010 to 31 December 2013. The observed and expected cases for this cluster were 178,296 and 153,004.1, respectively, with a likelihood ratio (observed/expected) of 1.2 and a *p*-value of <0.0001. The radius of this cluster was 95.5 km, encompassing northern Tigray and southern Eritrea (bordering Ethiopia).

The fifth probable cluster was also identified in the northwestern part of the country. The time frame for this particular cluster was 1 January 2017 to 31 December 2019. The likelihood ratio (observed/expected) was 8.5, with observed and expected cases of 1725 and 203.7, respectively, and a *p*-value of <0.0001. The radius of this cluster was 101.3 km, covering the central part of the Amhara region.

The sixth cluster was the smallest of all in terms of area coverage, with a radius of 36 km. The time frame of this cluster was from 1 January 2014 to 31 December 2016. The observed and expected cases for this cluster were 20.7, with a *p*-value of <0.0001. This cluster’s location bordered Afar and Djibouti.

The seventh and last cluster was found in north-central Amhara. The time frame for this cluster was from 1 January 2010 to 31 December 2013. The observed and expected cases were 29,954 and 25,859.1, respectively, with a likelihood ration of 1.2 and a *p*-value of <0.0001.

In this particular study, more than one cluster was found during the same timeframe. The first and the sixth clusters were found to be the largest (a radius of 380.9 km) and the smallest (a radius of 36.0 km) clusters, respectively, in terms of area coverage.

## 4. Discussions

Cross-sectional study designs for seroprevalence studies and purposive outbreak investigations are the routine study methods used in FMDV research in Ethiopia. However, there is a lack of retrospective analysis. Therefore, our study used retrospective outbreak data for this spatiotemporal outbreak distribution analysis. In this study, a total count of 704 outbreaks (an average 70.4 per year) and a sum of 1302 (an average 130.2 per year) FMD outbreak occurrences were observed. This is in agreement with the findings of Aman et al. [20], which recorded 35 FMD outbreaks in Amhara regional state only. In this study, it was seen that FMD outbreaks were reported from approximately 90% of the districts. The regional distribution was also seen to cover almost all regions, though the density of the outbreaks was high in the Oromia, Amhara, Tigray, and SNNP regions, with 443, 331, 255, and 192 outbreaks reported from 2010 to 2019, respectively. This is in agreement with the report that stated FMD outbreaks in Ethiopia are higher in Oromia and Amhara [33]. Our finding was again supported by the findings that reported higher FMD outbreaks in the central part of the country, mainly in the Oromia and Amhara regions [14,20,34].

This study also ascertained that in the last ten years, single outbreaks accounted for 84%, three to five outbreaks shared 12%, and more than five outbreaks shared 4% of the overall reports. This finding was comparable to that of Jemberu et al. [34], which identified that 73% of FMD outbreaks are single outbreaks. The epidemic cycle of FMD in Ethiopia over a 10 year period (from 2010 to 2019), as documented by this study, was five years in length. This contradicts with a report from the Amhara region which reported FMD epidemics every four years, and, combined with the other ten year retrospective studies of Asfaw and Sintaro in 1999 [35] and Ayelet in 2004 [14], suggests 3–4 year cyclic FMD outbreak patterns in Ethiopia [31]. This is also supported by other findings that reported three to six epidemic cycles in FMD endemic countries, with five year intervals [36,37,38,39,40]. In addition, all of the reported outbreaks were not confirmed due to inefficient sampling, diagnosis capacity, and outbreak investigation, and a lack of laboratory reagents.

High FMD occurrence was also associated with the dry season, extending from October to March. The dry season might be considered as the ideal time for the occurrence of an FMD outbreak. This period is the time of the year where animals are allowed to move from one place to another, freely, in search of pasture and water. Livestock marketing also affects livestock mobility. In the dry season of the year, when Christmas, Easter, and Epiphany festivities are celebrated, livestock demand is high. In this period, there is an increase in animals’ movements for sale. Moreover, this period of the year is also the crop harvest season, when crop remnants are grazed by free range livestock. Our findings are also in line with the findings of Aman et al. [20], which stated that the dry season, as well as religious festivals, are the determinants of FMD outbreaks in the Amhara region of Ethiopia.

The number of reported outbreaks through time from 2010 to 2019 only peaked in 2012 (357), 2017 (216), and 2019 (106). This variability between years might be attributed to a lack of reporting and inefficient outbreak communication. High outbreak numbers and low reported cases were seen in 2012 as compared to 2011, where the outbreak numbers were two-fold lower and the cases were much higher. In terms of monthly trends, the situation is the same across all the years, which starts in October, and the outbreaks drops from April onwards. This is in line with the finding of Lee et al. [41] in Vietnam, where FMD is also endemic.

In this study, it was found that only 12% of FMD outbreak samples were serotyped over the preceding ten year period. This low number of serotype identification might be attributed to a lack of outbreak communication, sampling, and reagents, as well as to limited laboratory facilities, where only the NAHDIC, Sebeta and NVI, and Debrezeit (Bishoftu) are capable of doing the FMD diagnostic tests (PCR and Antigen ELISA) at the national level.

This study is the first of its kind to evaluate the spatiotemporal patterns and space–time clusters of FMD at the national level using national outbreak data. In our study, seven clusters of FMD outbreak were found, covering most central highlands extending from the south to the northern parts of the country. This is an area where the high density of livestock production is practiced for commercial purposes. This is in agreement with the report of Leta and Mesele [42], which stated that this area is the most populous area in terms of livestock density. On the other hand, this area is also linked to good infrastructure (roads and markets), which allows for the free movement of animals, exacerbating the transmission of livestock diseases.

Though this study indicates the dynamism of FMD outbreaks using the temporal distribution, the serotype distribution, and a space–time cluster analysis, there are still limitations, including the under-reporting of outbreaks, tentative diagnosis reports based on clinical signs, and the use of centroid data. These may lead to misdiagnoses of the disease, as other animal diseases resemble FMD in their clinical presentation (e.g., swine vesicular diseases and bovine viral diarrhea virus) [43,44,45,46]. Therefore, this had an impact on our findings, as only 12% of the outbreaks are confirmed by laboratory tests. The centroid data used in this spatiotemporal analysis might not indicate the actual geographic coordinates. This is because clinical case identification and reporting are mainly done at the district level. Though this had an impact on the exact locations of the outbreaks, it can at least determine that the disease is a burden in that specific district. Therefore, there is a need to have regular outbreak investigations and serotype identification linked to a particular strategy for controlling the disease. The outbreak data used in this study indicate the districts from which they are reported.

## 5. Conclusions

FMD is among the endemic livestock diseases in Ethiopia, affecting almost all parts of the livestock rearing regions of the country. From the total sum of 1302 outbreak reports over ten years (2010 to 2019), single outbreak reports shared 84%, whereas greater than two outbreaks took 16% of the total share of overall reported outbreaks. In terms of district coverage, the majority of the districts experienced an outbreak at least once. It was observed that the temporal distribution of FMD in Ethiopia is seasonal (October to March), and the spatial distribution of the disease is found in the southern to northern parts of the country, following infrastructure development. The seven clusters identified here are also indicative of the distribution of FMD outbreaks in the country, mainly in this area. The serotypes identified in this period are O, A, SAT-2, and SAT-1; therefore prophylactic vaccination, movement restriction, and other bio-security measures should be applied at the beginning of and during the dry season. The time, space, and clusters of the disease identified in this study are important in identifying an area and time to be focused on in implementing any control, prevention, and prophylactic vaccination schedule. The seven clusters identified here are also indicative of where to implement these strategies. Outbreak timing and hotspot identification are the outputs of this study, and they are useful for policymakers. Moreover, it is important to raise stakeholder awareness on biosecurity management and, if possible, implement a national vaccination program before the dry season to prevent outbreaks and ongoing transmission.

## Figures and Tables

**Figure 1 viruses-14-01558-f001:**
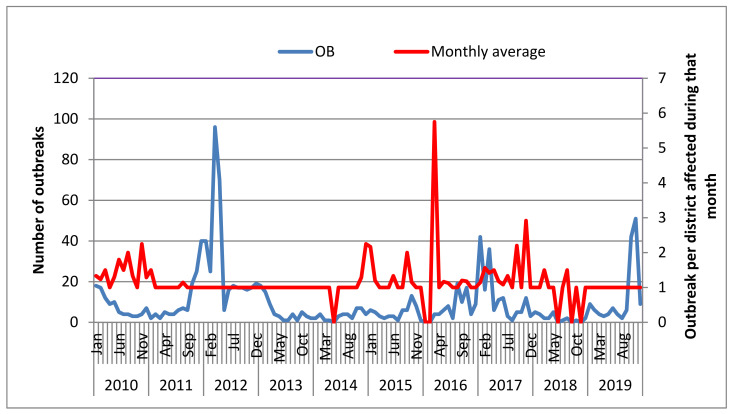
Outbreaks per district affected by one or more outbreaks during that month (indicated by the red line) and monthly outbreaks of FMD in Ethiopia over the period between 2010 and 2019. OB: outbreak (the number of outbreaks in each month is indicated by the blue line). Three peaks for outbreaks were observed in 2012, 2017, and 2019. Flat monthly averages, e.g., from September 2011 to March 2014, indicate that the number of affected districts is equal to the sum of the reported outbreaks.

**Figure 2 viruses-14-01558-f002:**
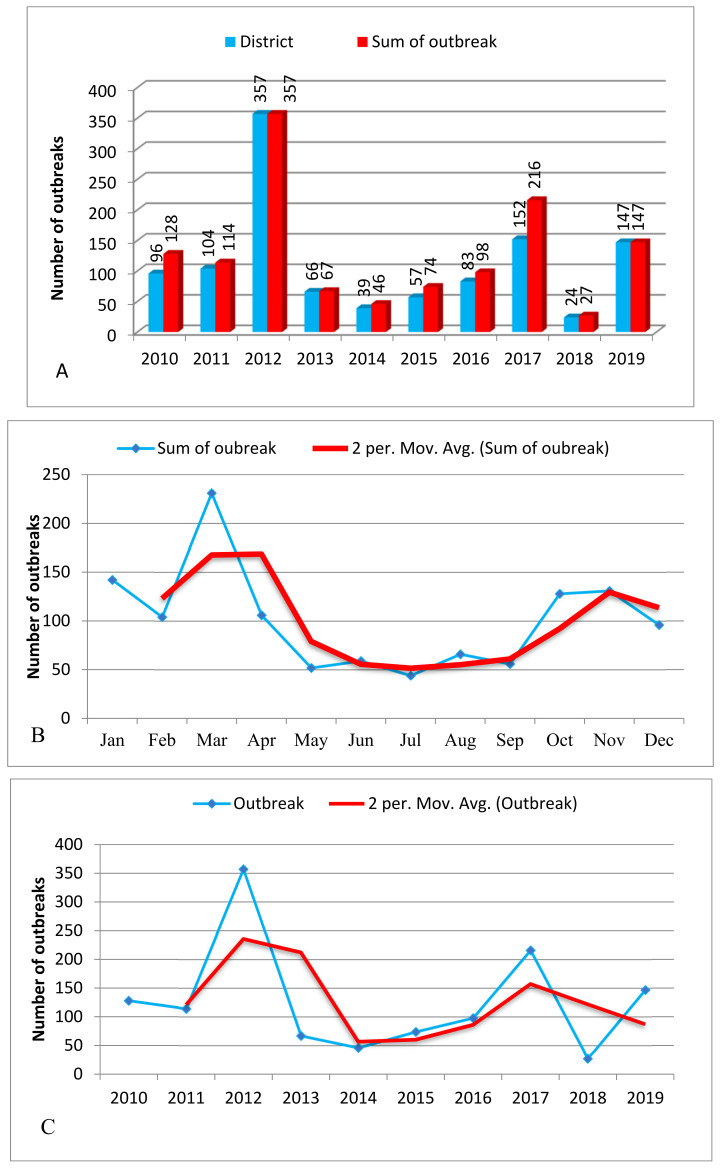
(**A**) Annual FMD outbreak in Ethiopia (2010–2019). The graph shows the number of districts affected in a specific year (grey bar) and the sum of the outbreaks in that specific year (black bar). (**B**) The monthly FMD outbreak distribution for ten years (2010–2019), with respective months (blue line) and its trend with months as a two-period moving average (red line) in Ethiopia. (**C**) Ten years of FMD outbreak distribution (blue line) and its trend as a two-period moving average (red line) in Ethiopia.

**Figure 3 viruses-14-01558-f003:**
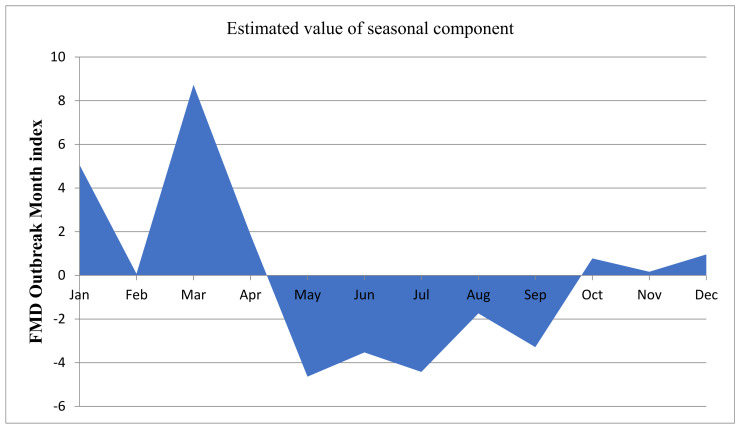
Seasonal indices (a measure of how a particular season through a cycle compares with the average season of that cycle) of monthly FMD outbreaks between 2010 and 2019 in Ethiopia. The positive values (upward wave) indicate that the trend component estimate does not have an impact on the estimate of the seasonal component, whereas the negative values indicate that the trend component affected the estimate of the seasonal component (downward wave). The figure shows the seasonality of FMD outbreak occurrences in Ethiopia.

**Figure 4 viruses-14-01558-f004:**
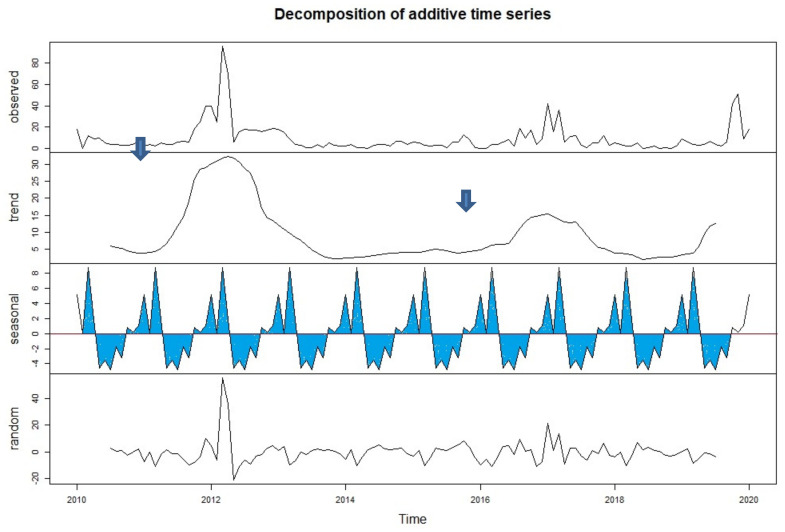
Decomposition of the time series of the observed temporal patterns of FMD outbreaks from 2010 to 2019. The graph shows the observed outbreak number (top panel) decomposed into three components (trend, seasonality, and random). Trend: the predicted outbreak time series based on the observed data; seasonality: patterns that repeat within a fixed time period (October to March); random: this is the residual of the original time series after the seasonal and trend time series are removed.

**Figure 5 viruses-14-01558-f005:**
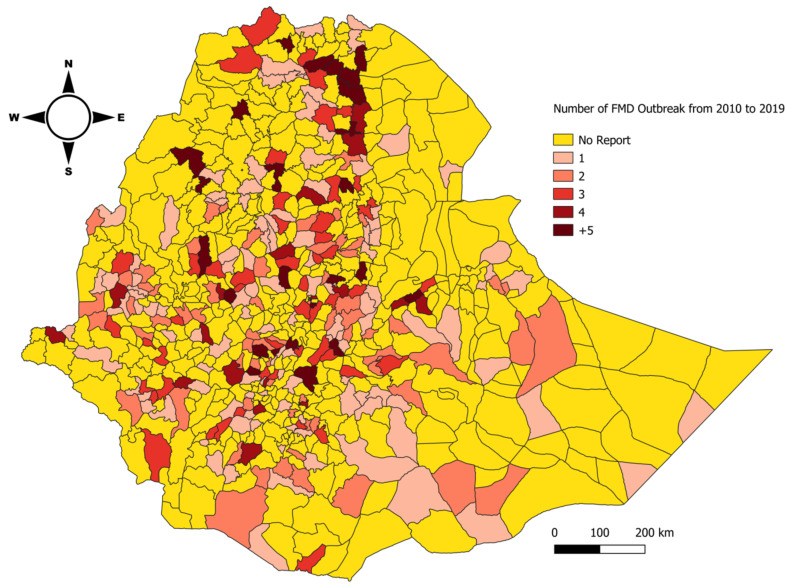
Map showing the Federal Democratic Republic of Ethiopia as of February 2022. The grey lines in the map indicate the district level administrative divisions. The shaded area shows district cumulative FMD outbreak reports in Ethiopia from 2010 to 2019.

**Figure 6 viruses-14-01558-f006:**
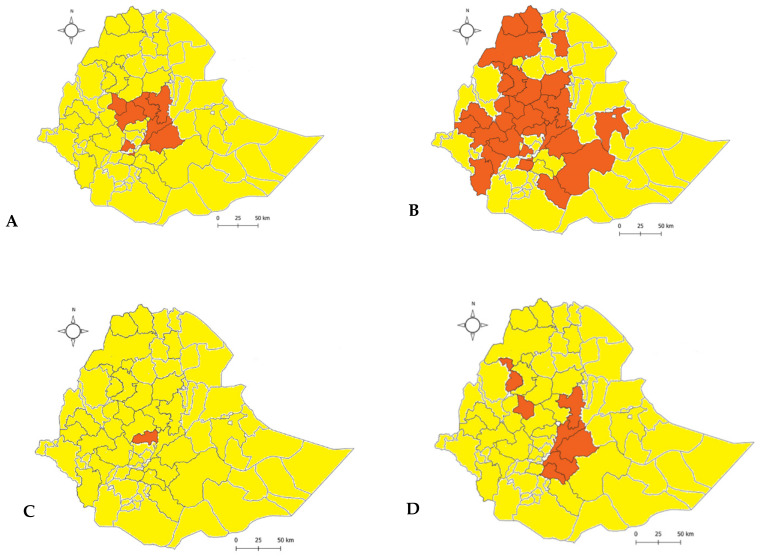
Identified FMD serotype distribution from 2010 to 2019 in Ethiopia. Panel (**A**): SAT-2, Panel (**B**): O, Panel (**C**): SAT-1, and Panel (**D**): A serotypes. This is the map of the Federal Democratic Republic of Ethiopia as of February 2022. The grey lines in the map indicate the district level administrative divisions. The red shaded regions are the FMDV serotype reports and their distributions.

**Figure 7 viruses-14-01558-f007:**
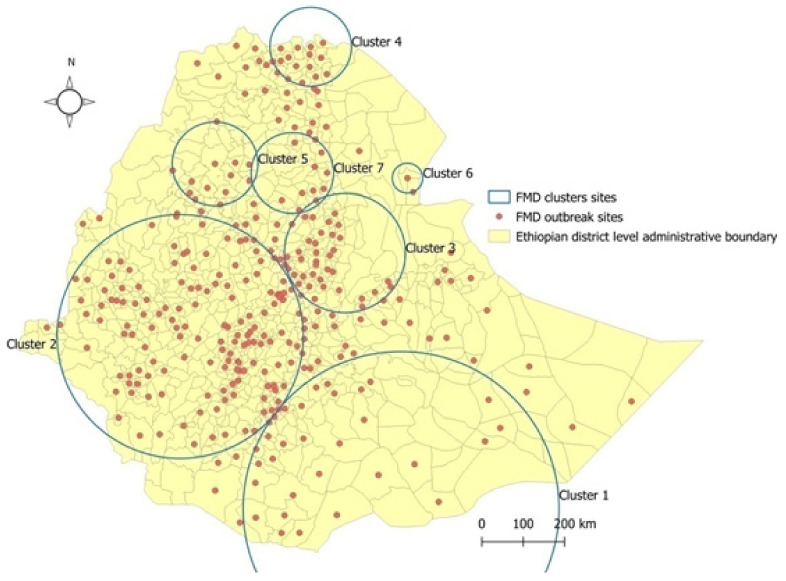
Map showing the Federal Democratic Republic of Ethiopia as of February 2022. The grey lines in the map indicate the district level administrative divisions. Included is the SaTScan cluster analysis of FMD outbreaks from 2010 to 2019 in Ethiopia (50% at risk). The red dots are outbreak spots and the light blue circles are clusters. Cluster 1: 1 January 2017 to 31 December 2019; Cluster 2: 1 January 2014 31 December 2016; Cluster 3: 1 January 2017 31 December 2019; Cluster 4: 1 January 2010 to 31 December 2013; Cluster 5: 1 January 2017 to 31 December 2019; Cluster 6: 1 January 2014 to 31 December 2016; and Cluster 7: 1 January 2010 to 31 December 2013.

**Table 1 viruses-14-01558-t001:** Livestock population by region (thousands) in Ethiopia.

Region	Cattle	Goat	Sheep	Region Share in %
Tigray	3119.40	3005.50	1388.10	7.4
Afar *	1627.20	3398.20	1799.00	6.8
Amhara	11,757.30	5468.60	9469.70	26.4
Oromiya	21,375.70	7678.40	9391.10	38.1
Somali *	675.6	1710.40	1316.80	3.7
Benishangul Gumuz	363.6	371.5	85.3	0.8
SNNP	9574.70	2624.60	4000.10	16
Gambella	212.6	54.6	48.1	0.3
Harari	40.8	41.2	5	0.1
Addis Ababa	89.5	19.1	21.8	0.1
Dire Dawa	49.8	154.7	59.6	0.3
Ethiopia	48,886.20	24,526.90	27,584.60	100

* Afar: only two of the five administrative zones and three of the nine administrative zones of Somali were included in annual surveys. SNNP: south nation nationality people; region share: the percentage of livestock population shared by the region.

**Table 2 viruses-14-01558-t002:** Yearly (1 January 2010 to 31 December 2019) outbreaks and cases of FMD in cattle and sheep.

Year	Species		Total
	Cattle	Sheep and Goats	
2010	22,440	20	22,460
2011	114,444	67,889	182,333
2012	94,344	736	95,080
2013	6722	194	6916
2014	4710	NR	4710
2015	25,578	NR	25,578
2016	7257	3	7260
2017	22,904	700	23,604
2018	1012	NR	1012
2019	7808	1	7809
Total	307,219	69,543	376,762

NR: not reported.

## Data Availability

The data presented in this study are available on request from the corresponding author. The data are publicly available.

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
