# Peer review of "Temporal and Spatial Patterns and a Space–Time Cluster Analysis of Foot-and-Mouth Disease Outbreaks in Ethiopia from 2010 to 2019"

_viruses, 2022, doi:10.3390/v14071558_

Round 1

Reviewer 1 Report

Comment to the author,

In this study, Woldemariyam, and colleagues have analysed the spatial and temporal distribution of FMD outbreaks in Ethiopia from 2010 to 2019 using national outbreak reports. For the analysis, they have used 376,762 cases and 1302 outbreaks from 704 districts based on the data obtained from the Minister of Agriculture of Ethiopia. As conclusions of the study, the authors have indicated the time, space, and cluster of the disease outbreak are important for identifying an area and time to be focused on in implementing any control, prevention, and prophylactic vaccination schedule. Moreover, the information/data analysed here will be a valuable summary for the policymakers in the country. I am pleased to accept this manuscript for publication in the Viruses. I hope readers from Ethiopia and the rest of the world will enjoy reading this study. As an adding point to the study, authors should precisely make specific conclusions based on the data analysed and include further remarks based on the currently available information; thus, government authorities will be able to gather essential data in their future surveys.

Reviewer 2 Report

This was an interesting manuscript aiming to describing spatio-temporal pattern of FMD outbreaks. Additionally, time series model is used to describe trend and seasonality of the data. However, several misconceptions in the analytical methods potential effects on the results and interpretation of the results. Misconceptions were found for time series decomposition analysis. Also, limitation in the parameter setting of spatio-temporal analysis results in less meaningful results (primary cluster is extremely large, with less meaningful). Some concepts on statistical analysis under spatio-temporal model are unclear. Methodology is not clear, for instance the location is centroid or actual location should be addressed since these two parameters offers different results. I would suggest to reject the manuscript as this form because of statistical and methodological misconceptions and numerous repetitions of results in the text and Table form, some important statements do not support by data or references, and the data visualization (all figures) is under standard.  Please see detail as follows.

  1. The introduction contents in abstract did not lead to the aim of the study. These contents are mostly about general knowledge on the FMD.
  2. The quality of Figure 2 is poor. Inconsistent with color using for visualization. Texts is too small and the alignment is out of good position (B) and improper enlargement of (C). A better visualization of the Figure 2 is recommended.
  3. Figure 3 was just from the method of crop and paste from another software. Unwanted part is shown in the Figure like small square box. The current form should be replaced with a better one.
  4. Line 185-186. The author stated that “The optimal temporal aggregate period was set for three years; longer or shorter durations”. Because the results are relied on this setting, this statement must be supported by some references. Or even better, the authors should show results that support this statement in the supplementary file.
  5. Line 260-262. The author stated that “This analysis was mainly to see whether the temporal distribution of outbreaks is not random by test of randomness. It was observed that the distribution is not random as compared to the observed outbreak reports”. It is a completely misunderstood concept. Time series decomposition does not do anything about testing of randomness. The technique is just decomposed actual data into trend, seasonality and reminder (noise) without any testes for the randomness. The random part shown in the Figure is just noise or white noise which is need to be tested by additional statistics which is not shown in the present study.
  6. line 318: “The likely hood ratio” should be replaced with “The likelihood ratio”.
  7. Numerous texts in space-time cluster analysis (line 305-345) are repeated information from Table 3. Also, no highlight found from the spatio-temporal model was stressed herein.
  8. Technically, the finding that radius of the primary cluster is extremely large (390.9km) may not provide any important information for the implication of the study. The SPT model need to be much improved. I recommend authors explore the following manuscripts to apply the setting of maximum report cluster size (Han et al., 2016; Lee et al., 2021) to the analysis to get a better meaningful result.

Han, J., Zhu, L., Kulldorff, M., Hostovich, S., Stinchcomb, D.G., Tatalovich, Z., Lewis, D.R., Feuer, E.J., 2016. Using Gini coefficient to determining optimal cluster reporting sizes for spatial scan statistics. International journal of health geographics 15, 1-11.

Lee, S., Moon, J., Jung, I., 2021. Optimizing the maximum reported cluster size in the spatial scan statistic for survival data. International Journal of Health Geographics 20, 1-14.

  1. Application of linear regression is not necessary for this study because the authors used time series data and the characteristic of the data is auto-correlation and thus linear regression is not appropriate for this data. I suggest authors to focus on the time series model and make a better explanation from such model. This study can be improved by using time series method to forecast the future outbreaks.

Reviewer 3 Report

The authors of this work examined the temporal, geographical, and space-time clustering of FMD epidemics in Ethiopia. Overall, it is worthy of publication.

However, the English language has to be enhanced. There are grammatical problems throughout, and the sentence structure in some areas is incorrect. Line 140, Lines 360–362, and 389, for example.

Line 49: FMD is found not only in Sub-Saharan Africa and South Asia, but also in other parts of Asia, the Middle East, and other parts of the world.

Line 54: instead of using the word under, use within.

Line 104: Whether it is possible to practise or is currently being practised

Lin124: There are ten administrative regions listed here, while there are eleven in Table 1.

Line 136-137: It's unclear what the authors are trying to say here.

Figure 1: it's clumsy. Make it readable and comprehendible.

Figure 2 A: To show the number of districts, I recommend using the secondary axis option.

Figures 1, 2, and 7 from the materials and methods section might easily be transferred to the findings section.

Line 192: Define "cases" in this context. If they are referring to specific animals, this must be stated clearly.

Only cases are given in Table 2. The outbreaks have not been reported.

Line 198: Please define the outbreaks in terms of space and time. What does an outbreak mean in this context?

Line 225: I believe it is Fig 2A.

Line 234: Please remove the bracket

Line 281: The statement is ambiguous.

Line 372-374 and 428-429:  It's unclear what they're trying to say here.

Author Response

Corrections to the Comments and suggestions of Reviewer 3

Q1: There are grammatical problems throughout, and the sentence structure in some areas is incorrect. Line 140, Lines 360–362, and 389, for example.

A1.1: Line 140: As per the reviewers suggestion the statement is corrected as follows: The count of an outbreak for ten years was also plotted separately and linear regression was applied to calculate the outbreak trend (Figure1).

A1:2: Line 360–362: As per the reviewers suggestion the statement is corrected as follows:  In this study a total count of 704 outbreaks (average 70.4/year) and a sum of 1302 outbreak (average130.2/year) FMD outbreak occurrences were observed.

A1:3: Line 389: As per the reviewers suggestion the statement is made clear as follows: Dry season of the year (where Christmas, Easter, and Epiphany festivity are celebrated, livestock demand is high. In period there is an increase in animals’ movements for sale.

Q2: Line 49: FMD is found not only in Sub-Saharan Africa and South Asia, but also in other parts of Asia, the Middle East, and other parts of the world.

A2: A per the reviewers suggestion the statement is corrected as follows: Foot-and-mouth disease (FMD) is a trans-boundary disease, endemic to Sub-Saharan Africa, South Asia, Middle East and other part of the world

Q3: Line 54: instead of using the word under, use within.

A3: As per the reviewers suggestion the word under is replaced by within: This virus is grouped within the family of Picornaviridae, genus Aphthovirus and has seven immunogenically distinct serotypes (O, A, C, SAT-1, SAT-2, SAT-3, and Asia1) [3,4].

Q4: Line 104: Whether it is possible to practise or is currently being practiced

A4:As per the reviewers suggestion the statement is corrected as follows:In the highlands besides crop-livestock production, semi-intensive and intensive livestock production systems with small to medium scale herds are practiced [22].

Q5: Lin124: There are ten administrative regions listed here, while there are eleven in Table 1

A5: The numerical corrections are made as follows: The Ethiopian administrative structure has frequently been subject to modification. As of January 2020 there are 9 administrative regions or states composed of 80 zones (administrative hierarchy between region and districts) and 2 administrative cities

Q6:Line 136-137: It's unclear what the authors are trying to say here.

A6: As per the reviewers suggestion the statement is corrected as follows: The average outbreaks per district per month were plotted by dividing the monthly outbreak sum during that specific month to a specific district.

Q7: Figure 1: it's clumsy. Make it readable and comprehendible.

A7: As per the reviewer suggestion Figure one was made to be readable and comprehendible

Q8: Figure 2 A: To show the number of districts, I recommend using the secondary axis option.

A8: As per the reviewer suggestion Figure 2 is changed to be more attractive showing both the district and the number of outbreak in different color.

Q9: Figures 1, 2, and 7 from the materials and methods section might easily be transferred to the findings section.

A9: Dear Reviewer these figure (Figure 1,2 and 7 ) are all in the result section

Q10: Line 192: Define "cases" in this context. If they are referring to specific animals, this must be stated clearly.

A10: A s per the reviewer suggestion   cases are defined and the statement is written as “A total of 376,762 cases (a specific animal with a suspected clinical sign of FMD) were reported over a time period between 2010 to 2019 from 704 districts of Ethiopia”

Q11: Line 198: Please define the outbreaks in terms of space and time. What does an outbreak mean in this context?

A11: As per the suggestion of the reviewer outbreak is defined as follows “In the last ten years, a total of 1302 outbreaks (two or more linked cases of the same illness or the situation where the observed number of cases exceeds the expected number or a single case of disease caused by a significant pathogen in a specific period) were reported from 704 out of 779 districts”.

Q12: Line 225: I believe it is Fig 2A.

A12: As per the reviewer the Figure 3.2 is corrected to figure 2A “Outbreaks were also summarized with respect to their reporting district in a specific year. As depicted in figure 2A, 357 districts in 2012, 152 districts in 2017 and 147 districts in 2019 reported an outbreak of FMD. On the other hand only 24 districts reported an outbreak in 2018”.

Q13: Line 234: Please remove the bracket

A13: As per the reviewers suggestion the bracket is removed.

Q14: Line 281: The statement is ambiguous.

A14: A s per the reviewer suggestion the statement was made clear as follows: ”Two, three, four or five outbreak accounted for 12% of the report”.

Q15: Line 372-374 and 428-429:  It's unclear what they're trying to say here.

A15:1: Line 372-374: As per the suggestion the statement is made clear as follows: This study also ascertained that in the last ten years single  outbreak accounted 84%, three to five outbreak shares 12% and more than five outbreak share 4% of the reports. This finding was comparable to Jemberu et al., [35] that identified 73% of FMD outbreak are single.

A15: 2: Line 428-429: As per the reviewers suggestions the statement is made clear as follows” FMD is among the endemic livestock diseases in Ethiopia affecting almost all parts of the livestock rearing regions of the country. From the total sum of 1302 outbreak reports in ten years (2010 to 2019), single outbreak report shares   84%, whereas greater than two outbreaks take 16% of the share”.

Round 2

Reviewer 2 Report

Technically, the manuscript has improved. There are some points need to be addressed and clarified. Some results need to be confirmed. Please see below:

1.       Line 184: The author should provide references for the publication that use centroid for the spatio-temporal analysis.

2.       The limitation of using centroids rather than actual coordinates should be discussed in the discussion part.

3.       The using of time series decompositions and linear regression to determine trend offered an inconsistent of data analysis. The time series decomposition method is appropriate for this data type (time series outbreak data). In contrast, linear regression assumptions require independent data and thus time series data is absolutely not independent data. When using both analyses, time series decomposition shows increased or decreased trends at some periods when examine overall study period. However, the linear regression shows only downward trend. This inconsistent may cause some confusions, especially statistical conceptual.

4.       It is very challenges that, in this study, the outbreak number was considered as normal distribution data (because the author used linear regression). I would suggest that the number of outbreaks is likely to be the data with Poisson distribution as they are count data not continuous data. Thus, fitting Poison data with linear regression is not appropriate. However, if the authors would like to confirm that the data are normally distributed. Please show results for assumption checking to demonstrate that number of outbreaks are normally distributed (e.g normal QQ plots of residuals). Also, if the author would like to include the linear regression in this paper, it should be acknowledged that the assumption of auto-correlation is ignored for the linear regression analysis.

Reviewer 3 Report

The manuscript can be accepted in the present form